# Consensus-Robust Transfer Attacks via Parameter and Representation Perturbations

**Shixin Li**[1], **Zewei Li**[1], **Xiaojing Ma**[1†], **Xiaofan Bai**[1], **Pingyi Hu**[1],
**Dongmei Zhang**[2], **Bin Benjamin Zhu**[2†]
[1]Huazhong University of Science and Technology     [2]Microsoft Corporation
[1]{shixinli, lizewei, lindahust, xiaofanbai, pingyihu}@hust.edu.cn
[2]{dongmeiz, binzhu}@microsoft.com

## Abstract

Adversarial examples crafted on one model often exhibit poor transferability to others, hindering their effectiveness in black-box settings. This limitation arises from two key factors: (i) *decision-boundary variation* across models and (ii) *representation drift* in feature space. We address these challenges through a new perspective that frames transferability for *untargeted attacks* as a *consensus-robust optimization* problem: adversarial perturbations should remain effective across a neighborhood of plausible target models. To model this uncertainty, we introduce two complementary perturbation channels: a *parameter channel*, capturing boundary shifts via weight perturbations, and a *representation channel*, addressing feature drift via stochastic blending of clean and adversarial activations. We then propose *CORTA* (COnsensus–Robust Transfer Attack), a lightweight attack instantiated from this robust formulation using two first-order strategies: (i) sensitivity regularization based on the squared Frobenius norm of logits' Jacobian with respect to weights, and (ii) Monte Carlo sampling for blended feature representations. Our theoretical analysis provides a certified lower bound linking these approximations to the robust objective. Extensive experiments on CIFAR-100 and ImageNet show that CORTA significantly outperforms state-of-the-art transfer-based methods—including ensemble approaches—across CNN and Vision Transformer targets. Notably, CORTA achieves a *19.1 percentage-point gain in transfer success rate over the best prior method* while using only a single surrogate model.

## 1 Introduction

Adversarial attacks [1, 2] pose a serious threat to deep neural networks (DNNs), as small, often imperceptible perturbations to input data can cause models to produce incorrect predictions. The risks are particularly severe in safety-critical domains such as facial recognition, autonomous driving, and medical diagnosis [3, 4]. Although numerous defenses have been proposed, *black-box* attacks remain especially concerning because they require no knowledge of the target model's architecture or parameters. Developing more effective black-box attacks is therefore critical for exposing model vulnerabilities and advancing robust evaluation practices.

Most black-box attacks rely on *transferability*: adversarial examples generated on a *surrogate* model are expected to fool an *unknown* target model [5]. Transferability has therefore become a focal point for exposing cross-model vulnerabilities. Recent progress spans multiple directions—gradient refinements [6, 7, 8, 9], input transformations [10, 11, 12, 13], model ensembles [14, 15, 16, 17], and feature-level objectives [18, 19, 20, 21, 22, 23]. While feature-level approaches begin to address

---

†Corresponding authors: *Xiaojing Ma (lindahust@hust.edu.cn)* and *Bin Benjamin Zhu (binzhu@microsoft.com)*.

39th Conference on Neural Information Processing Systems (NeurIPS 2025).

deeper causes of transfer failure—such as differences in internal representations—most existing methods do not explicitly or jointly tackle the full range of factors that limit transferability. Consequently, state-of-the-art success rates still drop sharply when the target's architecture diverges from the surrogate's, due to overfitting to surrogate-specific characteristics and limited generalization to unseen black-box models.

**Two Sources of Transfer Failure.** Our analysis identifies two independent factors that hinder adversarial transferability:

- *Decision-boundary variation.* The local classification boundary can shift significantly between the surrogate and target models due to differences in initialization, training procedures, or architecture. As a result, a sample located at the same position in decision space may still yield different predictions.

- *Representation drift.* The latent representations produced by different models for the same input can diverge, placing the sample at different locations in feature space. This shift can cause the input to fall on opposite sides of an otherwise similar decision boundary, leading to inconsistent outputs.

None of the existing attacks explicitly addresses both factors, leading to limited transferability.

**Consensus-Robust Approach.** We view transferability through a *consensus-robust* lens: any black-box target can be modeled as a perturbed version of a single surrogate, with uncertainty injected along two independent axes:

- *Parameter channel.* We model decision-boundary shifts—arising from differences in initialization, training, or architecture—as weight perturbations $\Delta W$ to the surrogate's parameters $W$. The perturbed model $f_{W+\Delta W}$ simulates local boundary variation between the surrogate and target models.

- *Representation channel.* Because an adversarial example resembles its clean counterpart, a target model—especially one with a different architecture—may preserve *clean* features that the surrogate suppresses. To emulate such variations, we blend the surrogate's clean and adversarial latent representations at selected layers, simulating feature-level deviations.

These two perturbation modes jointly define an *uncertainty set* $\mathcal{T}$ over plausible target models. We therefore pose transferability as a *set–robust* objective for *untargeted* attacks: choose input perturbations that *maximize the minimum* (worst–case) *loss* across $\mathcal{T}$—that is, raise the loss even for the *worst–case* target induced by *bounded parameter perturbations*, while taking expectation over *stochastic feature blending*. To make this tractable, we use two lightweight, first–order approximations: linearizing the *logits* with respect to parameter changes and Monte Carlo sampling to estimate the expectation over *feature blends*. Our theoretical analysis then provides a certified *lower bound* on this robust objective in terms of two estimable quantities—the *expected blended loss* and the *squared Frobenius norm of the logits' Jacobian with respect to parameters*—offering principled guarantees for *consensus–robust* transferability. We instantiate this formulation as *CORTA*, a query-free, optimizer-agnostic attack on a single surrogate that *maximizes* the *expected blended loss* while *regularizing* the *squared Frobenius norm of the logits' Jacobian*, jointly addressing representation drift and decision-boundary variation.

**Our Major Contributions:**

- *Consensus–robust formulation.* We frame black-box transferability as a *set–robust* objective for *untargeted* attacks: choose perturbations that *maximize the minimum loss* across an uncertainty set of plausible targets, induced by *bounded parameter perturbations* (decision-boundary variation) and *stochastic feature blending* (representation drift).

- *Dual-channel surrogate modeling.* We propose a unified transfer-based framework that emulates target variability via two channels on the surrogate: (i) the *parameter channel*, modeling boundary shifts through weight perturbations; and (ii) the *representation channel*, simulating feature-level discrepancies by blending clean and adversarial representations.

- *Principled first-order optimization with certified guarantees.* We develop lightweight, scalable approximations for each channel—*logit linearization* for parameter perturbations and *Monte Carlo* estimation for feature blends—and provide theoretical analysis yielding a certified *lower bound* on the robust target objective in terms of two estimable quantities: the *expected blended loss* and the *squared Frobenius norm of the logits' Jacobian with respect to parameters*.

- *Superior empirical results.* Our CORTA consistently surpasses state-of-the-art transfer-based black-box attacks—including ensemble-based methods—across diverse architectures (CNNs and ViTs) on ImageNet and CIFAR-100. For example, when transferring from ResNet-18 to Swin-B on CIFAR-100, CORTA achieves a 97.9% *transfer success rate*, outperforming *Ens*—the strongest existing method—by 19.1 percentage points (78.8%), while using only a *single surrogate model* (ResNet-18) compared to **Ens**'s ensemble of four surrogates (two CNNs and two ViTs).

## 2 Related Work

Transfer-based adversarial attacks seek to enhance the effectiveness of surrogate-generated examples on unseen models. Existing approaches can be grouped as follows:

- **Gradient-based refinements:** Momentum [6], advanced optimizers [7, 9, 24], and skip-gradient or linearized backpropagation [25, 26] aim to stabilize updates and escape local minima.
- **Input transformations:** Random resizing, translation, and image mixing (e.g., DI [10], TI [11], Admix [12]) increase input diversity to reduce overfitting to the surrogate.
- **Model ensembles:** These attacks improve transferability by ensembling surrogate models with diverse architectures [14, 15, 16]. To avoid training multiple models, *Ghost Networks* [27] and *LGV* [28] generate variants from a single network via dropout perturbation or high–learning-rate fine-tuning, while others ensemble checkpoints from one training trajectory [17]. Although these methods vary in how they introduce diversity, all require generating adversarial examples for multiple network instances, resulting in substantial computational overhead.
- **Feature-level attacks:** These methods manipulate intermediate representations to promote transferability. For example, TAP [18] and ILA [19] maximize the distance between clean and adversarial features at selected layers. FIA [20], BFA [29], and NAA [21] estimate feature importance using gradient-based attribution techniques. FPA [30] relies on permutation at a feature layer of a CNN-based surrogate model. CFM [22] mixes adversarial features with those of benign samples, while DHF [23] mixes adversarial and original features during attack generation.

Existing methods fail to directly address both decision-boundary variation and representation drift. Most inject diversity to bridge the surrogate–target gap without tackling these root causes. DHF is the closest to our approach on representation drift, but it remains heuristic and lacks a principled formulation. Consequently, these methods consistently underperform compared to our approach, which explicitly and jointly targets both sources of transferability failure.

## 3 Transferability as Consensus Robustness

### 3.1 Modeling Transferability via Parameter and Representation Perturbations

In Section 1 we argued that adversarial examples generated on a surrogate model $f_W$ may fail to transfer to an unseen target $f_\theta^t$ because of (i) *decision–boundary variation* and (ii) *representation drift*. We formalize these two factors as *parameter* and *representation* perturbations of the surrogate.

**Decision Boundary Variation as Parameter Perturbation.** Both $f_W$ and $f_\theta^t$ solve the same task, so their decision boundaries are generally similar despite differences in initialization, optimization, or even architecture [31, 32]. We model the target as a parameter-perturbed version of the surrogate:

$$f_\theta^t(x) \approx f_{W+\Delta W}(x), \tag{1}$$

where $\|\Delta W\|_F \leq \rho$ represents "nearby" models, and $x$ denotes an input with true label $y$. This gives an architecture-agnostic abstraction of decision-boundary shifts.

**Representation Drift as Feature Blending.** Adversarial examples usually remain visually and semantically similar to their originals, yet targets may retain features suppressed by the surrogate, causing intermediate mismatch. To capture this uncertainty, we stochastically blend, at a chosen set of layers S, the activations from the adversarial and clean inputs

$$z_\ell^{\text{blend}}(\lambda_\ell) = \lambda_\ell \, z_\ell^{\text{adv}} + (1 - \lambda_\ell) \, z_\ell^{\text{orig}}, \qquad \lambda_\ell \in [\lambda_{\min}, 1], \ \ell \in \mathcal{S}, \tag{2}$$

where $z_\ell^{\text{adv}} = z_\ell(x + \delta; W)$ and $z_\ell^{\text{orig}} = z_\ell(x; W)$ are computed at layer $\ell$ using the same weights $W$. This stochastic blending abstracts representation uncertainty across models.

**Robust Target Objective (Set–Robust Formulation).** We define the uncertainty set over targets as the product of bounded parameter and representation variations:

$$\mathcal{T} = \left\{ (\Delta W, \lambda) : \ \|\Delta W\|_F \leq \rho, \ \lambda \in [\lambda_{\min}, 1]^{|\mathcal{S}|} \right\}. \tag{3}$$

Maximizing *untargeted* transferability reduces to the consensus-robust optimization:

$$\max_{\delta \in \mathcal{B}_\epsilon} \ \min_{(\Delta W, \lambda) \in \mathcal{T}} \ \mathcal{L}\big(f_{W+\Delta W}(x + \delta; \{z_\ell^{\text{blend}}(\lambda_\ell)\}), \ y\big). \tag{4}$$

This "consensus–robust" objective requires an adversarial perturbation $\delta$ to induce high loss uniformly across nearby parameterizations and a range of representation blends.

## 3.2 Practical Approximations for Consensus Robustness

Direct optimization of Eq. (4) is intractable as $\mathcal{T}$ spans infinitely many perturbations. We approximate it via two first-order channels.

**Parameter Channel: Parameter Channel: Linearization.** For small $\Delta W$, a first–order Taylor expansion of the loss around $W$ yields

$$\mathcal{L}(f_{W+\Delta W}(\cdot), y) \approx \mathcal{L}(f_W(\cdot), y) + \nabla_W \mathcal{L}(f_W(\cdot), y) \cdot \Delta W, \tag{5}$$

where "$\cdot$" denotes the Frobenius inner product: $A \cdot B \triangleq \langle A, B \rangle_F = \text{tr}(A^\top B) = \sum_i A_i B_i$. Sensitivity to $\Delta W$ satisfies $\|\nabla_W \mathcal{L}\|_F \leq C_{\text{out}} \|J_W\|_F \leq \sqrt{2} \|\nabla_W f_W\|_F$, where $J_W = \nabla_W f_W$. Hence the worst-case linearized loss shift under $\|\Delta W\|_F \leq \rho$ is at most $C_{\text{out}} \|\nabla_W f_W\|_F \rho$. This identifies $\|\nabla_W f_W\|_F^2$ as a key quantity governing sensitivity to parameter perturbations and motivates its use as a regularization term in our practical formulation described later.

**Representation Channel: Monte Carlo Feature Blending.** We approximate robustness to representation drift by averaging the loss under random feature blends. At each step, for each $\ell \in \mathcal{S}$ we optionally enable blending and, if enabled, draw $\lambda_\ell$ uniformly from $[\lambda_{\min}, 1]$ and form $z_\ell^{\text{blend}}$ as in Eq. (2). This Monte Carlo procedure provides an unbiased estimator of the gradient of the consensus (expected) loss with respect to $\delta$, and will serve as a building block in our practical formulation introduced later.

## 3.3 Theoretical Analysis: A Lower–Bound Certificate for the Robust Target

We show that the approximation in Section 3.2 yields a computable *lower bound*—up to constants—on the robust target in Eq. (4), linking our practical surrogate to the original worst–case objective.

**Notation.** Let $\mathcal{L}_W(x, \delta; \lambda) := \mathcal{L}\big(f_W(x+\delta; \{z_\ell^{\text{blend}}(\lambda_\ell)\}), y\big)$ and denote the loss gradient with respect to parameters by $g_W(x, \delta; \lambda) := \nabla_W \mathcal{L}_W(x, \delta; \lambda)$. Let $J_W(x, \delta; \lambda) := \nabla_W f_W(x + \delta; \{z_\ell^{\text{blend}}(\lambda_\ell)\})$ be the Jacobian of the logits.

**Assumptions.** Fix $\varepsilon, \rho > 0$ and $\lambda_{\min} \in (0, 1)$. We assume: (i) Twice–differentiability in $W$ and a uniform Hessian spectral bound along $W \to W + \Delta W$: $\|H_{W^*}(x, \delta; \lambda)\|_2 \leq M$ for all $\delta \in \mathcal{B}_\varepsilon$, $\|\Delta W\|_F \leq \rho$, $\lambda \in [\lambda_{\min}, 1]^{|\mathcal{S}|}$, and some $W^* = W + \tau \Delta W$, $\tau \in (0, 1)$. (ii) For each blended layer $\ell \in \mathcal{S}$, the task loss is Lipschitz in the blended feature with constant $L_\ell^z$: $\big|\mathcal{L}_W(x, \delta; \lambda) - \mathcal{L}_W(x, \delta; \lambda')\big| \leq L_\ell^z \big\|z_\ell^{\text{blend}}(\lambda_\ell) - z_\ell^{\text{blend}}(\lambda_\ell')\big\|$ when $\lambda$ and $\lambda'$ differ only in coordinate $\ell$. (iii) The feature drift at layer $\ell$ is uniformly bounded for $\delta \in \mathcal{B}_\varepsilon$: $\big\|z_\ell(x + \delta; W) - z_\ell(x; W)\big\| \leq B_\ell(\varepsilon)$. A sufficient condition is that the layer mapping to $z_\ell$ is $\widehat{L}_\ell$–Lipschitz in the input, in which case $B_\ell(\varepsilon) \leq \widehat{L}_\ell \varepsilon$. (iv) The loss gradient with respect to logits is bounded: $\|\nabla_f \mathcal{L}(f, y)\| \leq C_{\text{out}}$ (e.g., $C_{\text{out}} \leq 2$ for cross–entropy with softmax).

**Parameter Channel: Lower Bound for the Min over $\Delta W$.** For any $\lambda$ and any $\delta \in \mathcal{B}_\varepsilon$, Taylor's theorem and Assumption (i) give, for all $\|\Delta W\|_F \leq \rho$,

$$\mathcal{L}_{W+\Delta W}(x, \delta; \lambda) \geq \mathcal{L}_W(x, \delta; \lambda) - \big\|g_W(x, \delta; \lambda)\big\|_F \|\Delta W\|_F - \tfrac{1}{2} M \|\Delta W\|_F^2.$$

Taking the minimum over $\|\Delta W\|_F \leq \rho$ and using Assumption (iv) and the chain rule, $\|g_W\| \leq C_{\text{out}} \|J_W\|$, yields

$$\min_{\|\Delta W\|_F \leq \rho} \mathcal{L}_{W+\Delta W}(x, \delta; \lambda) \geq \mathcal{L}_W(x, \delta; \lambda) - \rho \, C_{\text{out}} \big\|J_W(x, \delta; \lambda)\big\|_F - \tfrac{1}{2} M \rho^2. \tag{6}$$

**Representation Channel: Min–Mean Bound for Blending.** By Assumptions (ii)–(iii), the loss is Lipschitz in each $\lambda_\ell$ with constant $C_\ell := L_\ell^z B_\ell(\varepsilon)$, since $\|z_\ell^{\text{blend}}(\lambda_\ell) - z_\ell^{\text{blend}}(\lambda_\ell')\| = |\lambda_\ell - \lambda_\ell'| \, \|z_\ell^{\text{adv}} - z_\ell^{\text{orig}}\| \le |\lambda_\ell - \lambda_\ell'| \, B_\ell(\varepsilon)$. Hence, over the hypercube $\lambda \in [\lambda_{\min}, 1]^{|\mathcal{S}|}$ endowed with the $\ell_1$ metric,

$$\big| \mathcal{L}_W(x, \delta; \lambda) - \mathcal{L}_W(x, \delta; \lambda') \big| \le \sum_{\ell \in \mathcal{S}} C_\ell \, |\lambda_\ell - \lambda_\ell'|. \tag{7}$$

It follows that the range of $\mathcal{L}_W(x, \delta; \cdot)$ over $[\lambda_{\min}, 1]^{|\mathcal{S}|}$ is at most $(1 - \lambda_{\min}) \sum_{\ell \in \mathcal{S}} C_\ell$, and therefore, for any probability distribution $P_\lambda$ supported on $[\lambda_{\min}, 1]^{|\mathcal{S}|}$,

$$\min_{\lambda \in [\lambda_{\min}, 1]^{|\mathcal{S}|}} \mathcal{L}_W(x, \delta; \lambda) \ge \mathbb{E}_{\lambda \sim P_\lambda} \big[ \mathcal{L}_W(x, \delta; \lambda) \big] - (1 - \lambda_{\min}) \sum_{\ell \in \mathcal{S}} C_\ell. \tag{8}$$

In particular, this holds for the layer–wise Bernoulli–plus–uniform sampling scheme used by CORTA.

**Combined Certificate.** Combining Eqs. (6) and (8) and then taking expectation over $\lambda \sim P_\lambda$ yields, for any $\delta \in \mathcal{B}_\varepsilon$,

$$\min_{\substack{\|\Delta W\|_F \le \rho, \\ \lambda \in [\lambda_{\min}, 1]^{|\mathcal{S}|}}} \mathcal{L}\Big( f_{W + \Delta W}\big(x + \delta; \{z_\ell^{\text{blend}}(\lambda_\ell)\}\big), \, y \Big) \ge \mathbb{E}_{\lambda \sim P_\lambda} \Big[ \mathcal{L}\Big( f_W\big(x + \delta; \{z_\ell^{\text{blend}}(\lambda_\ell)\}\big), \, y \Big) \Big]$$

$$- \rho \, C_{\text{out}} \, \mathbb{E}_{\lambda \sim P_\lambda} \Big[ \big\| J_W(x, \delta; \lambda) \big\|_F \Big] - \tfrac{1}{2} M \rho^2 - (1 - \lambda_{\min}) \sum_{\ell \in \mathcal{S}} L_\ell^z B_\ell(\varepsilon). \tag{9}$$

Finally, by Jensen's inequality, $\mathbb{E}_\lambda \|J_W\|_F \le \sqrt{\mathbb{E}_\lambda \|J_W\|_F^2}$, so controlling the second moment of the Jacobian suffices to bound its expected norm; hence a squared–norm regularizer is a principled surrogate.

**Interpretation.** The bound in Eq. (9) shows that maximizing the expected blended loss and penalizing the Jacobian norm $\|\nabla_W f_W\|_F^2$ provably increases a certified lower bound on the true robust objective, up to additive terms that depend only on model smoothness, $\rho$, and blending/feature-drift constants.

## 4 COnsensus–Robust Transfer Attack (CORTA)

Guided by the lower-bound certificate in Eq. (9), CORTA constructs an input perturbation $\delta$ that simultaneously (i) forces the surrogate to misclassify, (ii) maintains a high loss under random representation blends, and (iii) limits sensitivity of the logits to parameter perturbations (decision-boundary variation). After presenting the optimization objective, we detail its practical realization—Parameter–Stability Regularization—followed by the iterative generation of adversarial examples.

### 4.1 Optimization Objective

Let $J_W(x + \delta) \equiv \nabla_W f_W(x + \delta)$ denote the logits' Jacobian with respect to parameters. For untargeted attacks[1], CORTA solves

$$\delta^\star = \arg\min_{\delta \in \mathcal{B}_\varepsilon} \Big\{ - \underbrace{\mathbb{E}_{\{\lambda_\ell \sim \mathcal{U}[\lambda_{\min}, 1]\}} \big[ \mathcal{L}_{\text{CE}}\big( f_W(x + \delta; \{z_\ell^{\text{blend}}\}), y \big) \big]}_{\text{representation channel}} + \beta \, \underbrace{\big\| J_W(x + \delta) \big\|_F^2}_{\text{parameter channel}} \Big\}, \tag{10}$$

where $\mathcal{U}$ denotes the uniform distribution and $\lambda_{\min} \in (0, 1)$ is a hyperparameter. The first term maximizes the expected cross-entropy loss under random feature blends (estimated via Monte Carlo), encouraging robustness to representation drift. The second term penalizes the squared Frobenius norm of the logits' Jacobian with respect to parameters, reducing sensitivity to decision-boundary variation. The trade-off coefficient $\beta > 0$ is tuned empirically.

---

[1]The targeted variant removes the negative sign in front of the expectation and replaces $\mathcal{L}_{\text{CE}}$ with the target-class loss.

## 4.2 Representation Channel: Stochastic Feature Blending

Let $\mathcal{S}$ be a set of latent layers whose activations are exposed for blending. During each attack iteration we perform the following Monte-Carlo procedure:

1. For every $\ell \in \mathcal{S}$, sample a Bernoulli variable $\tau_\ell \sim \mathrm{Bernoulli}(p_b)$ with blending probability $p_b \in [0, 1]$.

2. If $\tau_\ell = 1$, draw $\lambda_\ell \sim \mathcal{U}[\lambda_{\min}, 1]$ and mix the adversarial and clean activations:

$$z_\ell^{\mathrm{blend}} = \lambda_\ell \, z_\ell^{\mathrm{adv}} + (1 - \lambda_\ell) \, z_\ell^{\mathrm{orig}}; \qquad (11)$$

   otherwise set $z_\ell^{\mathrm{blend}} = z_\ell^{\mathrm{adv}}$.

The stochastic switch $\tau_\ell$ explores a neighborhood of possible representation drifts while preserving the adversarial signal when blending is disabled. As $p_b \to 0$ or $1$, CORTA reduces to ordinary PGD or full feature blending, respectively.

## 4.3 Adversarial Example Generation

Starting from a random initialization $\delta_0 \sim \mathcal{U}[-\varepsilon, \varepsilon]$, we refine the perturbation for $T$ iterations. For clarity we present the basic I-FGSM update, but any gradient-based refinement, such as MI-FGSM [6] or NI-FGSM [7], can be plugged in unchanged, as CORTA is optimizer-agnostic.

$$\delta_{i+1} = \mathrm{clip}_\varepsilon \Big( \delta_i + \alpha \, \mathrm{sign}\big(\nabla_{\delta_i} \big[ \mathcal{L}_{\mathrm{CE}}\big(f_W(x + \delta_i; \{z_\ell^{\mathrm{blend}}\}), y\big) - \beta \, \|J_W(x + \delta_i)\|_F^2 \big]\big)\Big), \quad (12)$$

where $\alpha$ is the step size and $\mathrm{clip}_\varepsilon$ projects the perturbation onto the $\ell_\infty$ ball of radius $\varepsilon$ centered at $x$. The blended features in Eq. (11) are recomputed at each iteration, so the optimization implicitly minimizes the expectation in Eq. (10). By jointly penalizing parameter sensitivity and injecting stochastic feature blending, CORTA generates adversarial examples that transfer reliably across diverse architectures and training procedures.

# 5 Experiments

## 5.1 Experimental Setting

**Datasets.** We follow [16] and evaluate on two benchmarks: an ImageNet-compatible dataset[2] and CIFAR-100 [33]. All reported results are averaged over the entire ImageNet-compatible dataset and the full CIFAR-100 test set.

**Models.** *Target models:* We use diverse architectures, including CNNs (ResNet-50 [34], WideResNet-101 [35], BiT-M-R50 [36], BiT-M-R101 [36]) and vision transformers (ViT-Base [37], DeiT-Base [38], Swin-Base [39], Swin-Small [39]).

*Surrogate models:* For both CORTA and other non-ensemble baselines, we use a single surrogate model—ResNet-18 for CNN-based attacks and ViT-Tiny for ViT-based attacks. For checkpoint-ensemble attacks, we follow [17], which also adopts a single surrogate (either ResNet-18 or ViT-Tiny) but aggregates multiple checkpoints from the same model architecture. For ensemble-based attacks such as Ens and AdaEA, we follow [40] and adopt a multi-architecture surrogate setup comprising four models: ResNet-18, Inception-v3, ViT-Tiny, and DeiT-Tiny. All models are pretrained and obtained from PyTorch Image Models [41].

**Attack Baselines.** We compare against strong transfer-based black-box attacks: ensemble-based (Ens [14], AdaEA [16], Checkpoints [17]), feature-level (DHF [23], BFA [29]), input transformation (Admix [12]), and gradient-based (ANDA [24]). Official code and default settings are used unless otherwise specified.

**Defenses.** We evaluate CORTA against adversarial training [42, 43] and input transformation-based defenses, including JPEG compression [44], Randomized Resizing and Padding (R&P) [45], Bit

---

[2]https://github.com/cleverhans-lab/cleverhans/tree/master/cleverhans_v3.1.0/examples/nips17_adversarial_competition/dataset

Table 1: TSRs (%) on CIFAR-100 and ImageNet with I-FGSM, using ResNet-18 as the surrogate except for Ens and AdaEA, which use 2 CNN and 2 ViT surrogates. **Bold** indicates best performance.

| Dataset | Attack | CNN | | | | | ViT | | | | |
|---|---|---|---|---|---|---|---|---|---|---|---|
| | | RN-50 | WRN-101 | BiT-50 | BiT-101 | Avg. | ViT-B | Deit-B | Swin-B | Swin-S | Avg. |
| CIFAR-100 | Admix | 81.5 | 88.6 | 72.4 | 72.4 | 78.7 | 31.3 | 33.0 | 41.6 | 57.1 | 40.8 |
| | Ens | 92.0 | 87.4 | 83.3 | 73.3 | 84.0 | 75.2 | 89.1 | 78.8 | 85.4 | 82.1 |
| | AdaEA | 86.3 | 82.6 | 76.3 | 67.0 | 78.1 | 64.0 | 79.0 | 68.7 | 81.3 | 73.3 |
| | DHF | 90.2 | 92.3 | 79.1 | 75.0 | 84.1 | 39.3 | 33.7 | 36.5 | 57.9 | 41.9 |
| | BFA | 89.9 | 92.5 | 77.0 | 73.2 | 83.2 | 40.6 | 36.5 | 39.9 | 58.5 | 43.9 |
| | ANDA | 88.6 | 94.2 | 77.9 | 77.6 | 84.6 | 40.3 | 39.5 | 44.6 | 58.3 | 45.7 |
| | Checkpoints | 90.3 | 97.7 | 94.7 | 91.3 | 93.5 | 64.0 | 61.7 | 54.7 | 73.0 | 63.4 |
| | **Ours** | **97.3** | **98.8** | **98.2** | **96.2** | **97.6** | **96.8** | **93.5** | **97.9** | **97.1** | **96.3** |
| ImageNet | Admix | 91.9 | 83.6 | 79.5 | 71.1 | 81.5 | 26.4 | 38.6 | 29.6 | 36.3 | 32.7 |
| | Ens | 71.2 | 63.2 | 62.5 | 54.9 | 63.0 | 42.9 | 62.9 | 26.6 | 36.6 | 42.3 |
| | AdaEA | 73.5 | 61.4 | 59.1 | 50.9 | 61.2 | 36.9 | 53.8 | 25.0 | 33.4 | 37.3 |
| | DHF | 96.8 | 92.8 | 90.8 | 84.6 | 91.3 | 37.8 | 51.3 | 43.5 | 52.6 | 46.3 |
| | BFA | 97.9 | 95.8 | 93.6 | 89.6 | 94.2 | 43.0 | 53.2 | 47.8 | 57.3 | 50.3 |
| | ANDA | 94.4 | 86.1 | 81.7 | 73.3 | 83.9 | 36.8 | 52.2 | 38.8 | 47.0 | 43.7 |
| | Checkpoints | 95.5 | 95.7 | 95.9 | 90.6 | 94.4 | 45.1 | 56.0 | 40.4 | 51.2 | 48.2 |
| | **Ours** | **98.5** | **95.8** | **95.5** | **92.4** | **95.5** | **47.6** | **63.8** | **54.2** | **64.1** | **57.4** |

Table 2: TSRs (%) on ImageNet with I-FGSM, using ViT-Tiny as the surrogate, except for Ens and AdaEA, which use 2 CNN and 2 ViT surrogates. **Bold** indicates best performance.

| Attack | CNN | | | | | ViT | | | | |
|---|---|---|---|---|---|---|---|---|---|---|
| | RN-50 | WRN-101 | BiT-50 | BiT-101 | Avg. | ViT-B | Deit-B | Swin-B | Swin-S | Avg. |
| Admix | 37.7 | 43.9 | 49.7 | 42.7 | 43.5 | 48.5 | 63.9 | 26.6 | 31.9 | 42.7 |
| Ens | 71.2 | 63.2 | 62.5 | 54.9 | 63.0 | 42.9 | 62.9 | 26.6 | 36.6 | 42.3 |
| AdaEA | **73.5** | 61.4 | 59.1 | 50.9 | 61.2 | 36.9 | 53.8 | 25.0 | 33.4 | 37.3 |
| DHF | 46.0 | 52.4 | 55.6 | 49.4 | 50.9 | 64.0 | 76.5 | 32.9 | 40.8 | 53.6 |
| BFA | 51.1 | 56.3 | 59.5 | 52.8 | 54.9 | 72.9 | 85.8 | 34.5 | 44.9 | 59.5 |
| ANDA | 58.4 | 65.5 | 68.5 | 62.0 | 63.6 | 65.0 | 74.3 | 39.7 | 47.4 | 56.6 |
| Checkpoints | 41.0 | 42.9 | 49.8 | 41.0 | 43.7 | 54.9 | 81.9 | 26.7 | 34.9 | 49.6 |
| **Ours** | 63.6 | **70.4** | **74.4** | **68.3** | **69.2** | **77.8** | **87.8** | **50.7** | **58.4** | **68.7** |

Depth Reduction (Bit-R) [46], Feature Distillation (FD) [47], and Neural Representation Purifier (NRP) [48].

**Evaluation Metrics.** We report *Transfer Success Rate (TSR)*: the attack success rate on the target model for adversarial examples that are misclassified by the surrogate.

**Implementation Details.** All attacks are untargeted and evaluated under an $L_\infty$ bound of $\epsilon = 16/255$ for $T = 100$ iterations with a step size of $\alpha = 1.6/255$. The regularization weight is set to $\beta = 0.1$, chosen to balance the magnitudes of the two loss terms in Eq. 10 on the surrogate model. The blending probability is set to $p_b = 0.5$ based on surrogate optimization performance, and the blending proportion $\lambda$ is sampled from $\mathcal{U}[0.25, 1]$ to ensure sufficient feature mixing without reducing generation success. Stochastic feature blending is applied to all layers for CNN surrogates and to all linear layers for ViT surrogates. I-FGSM is used as the default method for generating adversarial examples. All experiments are implemented in PyTorch and conducted on two NVIDIA RTX 3090 GPUs.

## 5.2 Adversarial Transferability

**Attack on Standard Target Models.** Table 1 compares CORTA and baselines on CIFAR-100 and ImageNet across CNN and ViT targets, using *ResNet-18* as the surrogate for all methods except ensemble-based Ens and AdaEA, which use two CNN plus two ViT surrogates (see Section 5.1).

CORTA achieves the highest TSRs for every target model, across both CNN and ViT targets. On CNN targets, it attains average TSRs of 97.6% (CIFAR-100) and 95.5% (ImageNet), outperforming the best baselines by 13.0% and 1.1%, respectively. On ViT targets, despite using only a single

CNN surrogate, CORTA achieves 96.3% (CIFAR-100) and 57.4% (ImageNet), surpassing even the ensemble-based methods—which utilize ViT surrogates—by 14.2% and 7.1%. Notably, when transferring from ResNet-18 to Swin-B on CIFAR-100, CORTA achieves a 19.1% higher TSR (97.9% vs. 78.8%) compared to ensemble-based Ens, the strongest baseline.

These results demonstrate CORTA's strong transferability across datasets and model families. As most prior work focuses on ImageNet [12, 23, 24], subsequent experiments primarily report ImageNet results for consistency.

Table 2 reports TSRs on ImageNet using *ViT-Tiny* as the surrogate for all methods, except for ensemble-based Ens and AdaEA, which continue to use two CNN plus two ViT surrogates. CORTA consistently achieves the best overall performance, with average TSR gains of 5.6% on CNN targets and 9.2% on ViT targets over the strongest alternatives. Furthermore, CORTA outperforms all baselines on nearly every individual target model, with the only exception being RN-50, where it remains competitive and is surpassed only by the ensemble-based Ens and AdaEA. These results underscore CORTA's robustness and effectiveness across different surrogate architectures. We also report error bars in Appendix A.

Table 3: TSRs (%) on ImageNet with I-FGSM against various defenses, using ResNet-18 as surrogate except for Ens and AdaEA (2 CNN and 2 ViT surrogates). **Bold** indicates best performance. **Left**: TSRs for adversarially trained models. **Right**: average TSRs for input transformation defenses.

| Attack | Adversarial Training Defense | | | | Input Transformation-Based Defenses | | | | | |
|---|---|---|---|---|---|---|---|---|---|---|
| | Inc-v3ens3 | Inc-v3ens4 | Inc-v2ens | Avg. | R&P | Bit-R | JPEG | NRP | FD | Avg. |
| Admix | 54.1 | 52.6 | 38.9 | 48.5 | 59.2 | 56.5 | 52.1 | 25.2 | 57.8 | 50.2 |
| Ens | 37.3 | 36.0 | 22.7 | 32.0 | 48.3 | 46.6 | 43.6 | 24.4 | 50.7 | 42.7 |
| AdaEA | 30.6 | 30.1 | 19.4 | 26.7 | 45.5 | 50.4 | 39.4 | 23.9 | 47.0 | 41.2 |
| DHF | 63.3 | 60.6 | 45.4 | 56.4 | 70.5 | 68.1 | 58.3 | 28.4 | 68.2 | 58.7 |
| BFA | 69.3 | 62.8 | 49.4 | 60.5 | 76.2 | 72.0 | 62.7 | 33.1 | 72.6 | 63.3 |
| ANDA | 55.7 | 53.2 | 39.5 | 49.5 | 67.6 | 62.6 | 58.2 | 26.0 | 63.9 | 55.7 |
| Checkpoints | 73.9 | 72.4 | 57.4 | 67.9 | 76.4 | 71.8 | 69.3 | 28.9 | 72.2 | 63.7 |
| **Ours** | **76.5** | **72.8** | **60.1** | **69.8** | **78.1** | **75.6** | **70.8** | **41.2** | **76.3** | **68.4** |

Table 4: CORTA's TSRs (%) on ImageNet with ResNet-18 as surrogate, using different adversarial example generation methods. $\Delta$ indicates improvement over I-FGSM.

| Base | CNN | | | | | ViT | | | | |
|---|---|---|---|---|---|---|---|---|---|---|
| | RN-50 | WRN-101 | BiT-50 | BiT-101 | Avg. ($\Delta$) | ViT-B | DeiT- B | Swin-B | Swin-S | Avg. ($\Delta$) |
| I-FGSM | 98.5 | 95.8 | 95.5 | 92.4 | 95.5 | 47.6 | 63.8 | 54.2 | 64.1 | 57.4 |
| MI-FGSM | 98.7 | 96.3 | 95.7 | 93.3 | 96.0 (+0.5) | 52.7 | 68.2 | 56.8 | 67.0 | 61.2 (+3.8) |
| DIM-FGSM | 98.8 | 96.8 | 96.7 | 95.9 | 97.0 (+1.5) | 67.1 | 80.4 | 71.0 | 79.0 | 74.3 (+16.9) |

**Attack on Defended Target Models.** To further assess practical effectiveness, we evaluate all methods against two categories of defenses: (i) *adversarial training*, using three adversarially trained models, and (ii) *input transformation-based defenses*. The results are summarized in Table 3.

CORTA achieves the highest TSRs across both defense types. Specifically, it obtains an average TSR of 69.8% against adversarially trained models and 68.4% against input transformation defenses. In comparison to the strongest baseline methods, these results represent improvements of 1.9% and 4.7%, respectively.

These gains demonstrate that CORTA not only transfers effectively under standard conditions but also maintains strong robustness against advanced defense strategies.

Table 5: Generation success rates on ImageNet with ResNet-18 surrogate.

| Dataset | Admix | Ens | AdaEA | DHF | BFA | ANDA | Checkpoints | **Ours** |
|---|---|---|---|---|---|---|---|---|
| ImageNet | 97.0 | 100 | 100 | 99.4 | 99.4 | 96.3 | 100 | 69.9 |

Table 6: Computation time (s) per adversarial sample on ImageNet with ResNet-18 surrogate.

| Dataset | Admix | Ens | AdaEA | DHF | BFA | ANDA | Checkpoints | **Ours** |
|---------|-------|-----|-------|-----|-----|------|-------------|----------|
| ImageNet | 1.3 | 5.2 | 18.8 | 2.0 | 1.8 | 2.1 | 17.2 | 1.7 |

**Generating Adversarial Examples with Advanced Strategies.** CORTA uses I-FGSM as the default method, but it is compatible with stronger adversarial example generation methods. We compare I-FGSM with two enhanced variants: MI-FGSM [6] and DI [10], on ImageNet with ResNet-18 as the surrogate. As shown in Table 4, integrating MI-FGSM improves average TSRs from 95.5% to 96.0% on CNN targets and from 57.4% to 61.2% on ViT targets. Combining MI-FGSM with input diversity (DIM-FGSM) further boosts transferability, achieving 97.0% on CNN and 74.3% on ViT targets—absolute gains of 1.5% and 16.9%, respectively. These results demonstrate that CORTA benefits from stronger gradient-based attacks, further enhancing transferability across both CNN and ViT models.

## 5.3 Generation Success Rate on Surrogate

In addition to the TSRs reported above, we evaluate the *generation success rate* (GSR), defined as the proportion of adversarial examples that successfully mislead the *surrogate model* during attack generation. Table 5 presents the results: most baselines (Ens, AdaEA, DHF, BFA, Checkpoint) achieve nearly 100% success, whereas CORTA attains a lower rate of 69.9%.

**Why is CORTA's surrogate success lower?** This reduction mainly stems from two factors intrinsic to its set-robust formulation:

1. *Dual-objective optimization.* Unlike most baselines that optimize a single loss, CORTA jointly optimizes two objectives—representation and parameter channels—making the optimization problem more challenging.
2. *Feature blending interference.* The feature blending operation integrates the original sample's latent features, which can partially conflict with the adversarial perturbation objective, reducing surrogate success rates.

**Is this lower surrogate success a practical problem?** Not really—the key metric is transfer success rate (TSR), not surrogate success. In practice, attackers can simply discard unsuccessful examples on the surrogate and retain only those that succeed. This adds only modest overhead: generating the same number of successful examples requires optimizing about 1.43 times more samples (e.g., 100/69.9 compared to attacks achieving 100% surrogate success).

## 5.4 Computational Cost

Beyond transfer success rates and surrogate generation success rates, computational efficiency is also crucial. Table 6 reports the average time to generate an adversarial example. CORTA, requiring only a single surrogate, matches the speed of other single-model attacks and is significantly faster than ensemble-based methods. Thus, CORTA achieves superior transferability without additional computational overhead.

## 5.5 Ablation Study

**CORTA Components.** We evaluate the contributions of Parameter–Stability Regularization and Stochastic Feature Blending by comparing CORTA with: both components, each component individually, and neither (i.e., standard I-FGSM). Table 7 shows that both components are essential, with the best TSRs achieved when combined.

**Hyperparameter Sensitivity.** We assess the impact of three hyperparameters—parameter stability weight $\beta$, blending lower bound $\lambda_{min}$, and blending probability $p_b$—by generating adversarial examples on ImageNet (ResNet-18 surrogate) and evaluating TSRs on CNN and ViT targets. As shown in Figs. 1–3, CORTA maintains stable performance for $\beta$ in $[0.01, 0.1]$, $p_b$ in $[0.5, 1]$, and $\lambda_{min}$ in $[0.1, 0.3]$. These results indicate moderate sensitivity and robust performance across a range of hyperparameter values.

Table 7: Impact of CORTA components on ImageNet (ResNet-18 surrogate). ●: used; ○: absent. **Bold** indicates best performance.

| Ablation | | CNN | | | | | ViT | | | | |
| --- | --- | --- | --- | --- | --- | --- | --- | --- | --- | --- | --- |
| Representation | Parameter | RN-50 | WRN-101 | BiT-50 | BiT-101 | Avg. | ViT-B | Deit-B | Swin-B | Swin-S | Avg. |
| ○ | ○ | 61.1 | 48.2 | 41.7 | 33.3 | 46.1 | 10.8 | 16.5 | 12.9 | 15.7 | 14.0 |
| ○ | ● | 65.3 | 58.2 | 46.4 | 39.0 | 52.2 | 12.1 | 19.3 | 14.1 | 15.4 | 15.2 |
| ● | ○ | 97.1 | 92.8 | 91.3 | 85.0 | 91.5 | 44.6 | 58.0 | 48.3 | 58.4 | 52.3 |
| ● | ● | **98.5** | **95.8** | **95.5** | **92.4** | **95.5** | **47.6** | **63.8** | **54.2** | **64.1** | **57.4** |

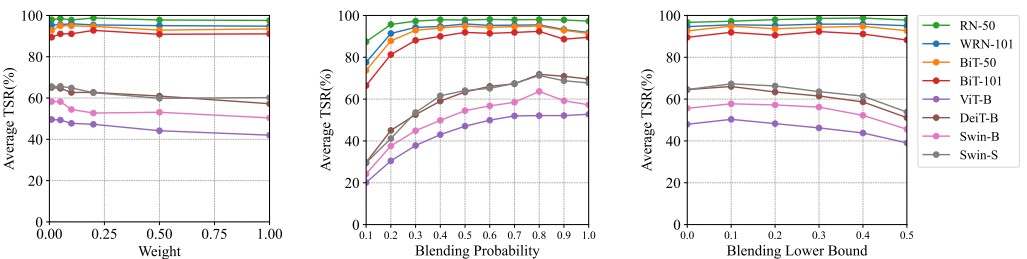

Figure 1: Weight ($\beta$).   Figure 2: Blend probability ($p_b$). Figure 3: Blend lower bound ($\lambda_{min}$).

# 6   Limitation

A potential limitation of our method is the need to compute second-order derivatives for each sample independently during backpropagation. Although frameworks like PyTorch and TensorFlow support automatic differentiation, their second-order computations typically aggregate curvature across a batch rather than compute per-sample values, limiting batch parallelization and increasing computational overhead. Nevertheless, as shown in Section 5.4, CORTA's single-surrogate optimization keeps the overall time cost practical.

Another limitation is that optimizing adversarial examples on the surrogate model with CORTA can be more challenging than with other single-model methods. For example, on ImageNet, CORTA achieves a generation success rate of 69.9%, compared to 96.3% for ANDA. However, this reflects performance on the surrogate model, while our goal is to generate adversarial examples that successfully attack the target model. As long as the generation cost on the surrogate is low and the resulting examples are effective against the target model, a reasonably lower generation success rate on the surrogate model does not diminish the practical effectiveness of our approach.

# 7   Conclusion

We introduced a consensus-robust framework for transfer-based *untargeted* adversarial attacks, explicitly addressing two underexplored factors limiting transferability: *decision-boundary variation* and *feature representation drift*. Our formulation models a neighborhood of plausible targets through *parameter perturbations* and *representation blending*, leading to a principled set-robust objective tailored for untargeted transfer attacks.

To make this objective tractable, we proposed two scalable first-order approximations with theoretical guarantees and instantiated them as *CORTA*, an efficient attack requiring only a single surrogate. CORTA integrates sensitivity regularization with stochastic feature blending, enabling attacks that are significantly more transferable across model families and training variations.

Extensive experiments on CIFAR-100 and ImageNet show that CORTA surpasses both single-surrogate and ensemble-based attacks while requiring only one surrogate model. For example, on CIFAR-100, transferring from ResNet-18 to Swin-B, CORTA achieves a *97.9% TSR*, exceeding the strongest baseline by *19.1 points* despite using far fewer resources.

Looking ahead, a key direction is extending this framework to *targeted attacks*, which imposes much stricter requirements: crafting perturbations that not only transfer but also steer predictions toward a specific target class under significant model variability. This extension poses a challenging but critical step toward building comprehensive evaluations of adversarial robustness.

## Acknowledgements

This work was supported in part by the National Natural Science Foundation of China (Grant No. 62272175), the Major Research Plan of Hubei Province (Grant/Award No. 2023BAA027), the Key Research & Development Plan of Hubei Province of China (Grant No. 2024BAB049), and the project of Science, Technology and Innovation Commission of Shenzhen Municipality of China (Grant No. GJHZ20240218114659027).

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

# A    Error Bars

Reporting confidence intervals offers greater transparency into the reliability of experimental results, particularly given the inherent randomness in adversarial attack evaluations. To capture this variability, we repeated the experiments from Table 1 and Table 2 on 100 randomly selected samples, running each setting 20 times with different random seeds. All reported values are presented as mean $\pm$ standard deviation. The corresponding results with error bars are summarized in Table 8 and Table 9.

Table 8: Mean $\pm$ standard deviation of TSRs (%) on CIFAR-100 and ImageNet with I-FGSM, using ResNet-18 as the surrogate except for Ens and AdaEA, which use 2 CNN and 2 ViT surrogates.

| Dataset | Attack | CNN Models | | | | | ViT Models | | | | |
|---|---|---|---|---|---|---|---|---|---|---|---|
| | | RN-50 | WRN-101 | BiT-50 | BiT-101 | Avg. | ViT-B | DeiT-B | Swin-B | Swin-S | Avg. |
| CIFAR-100 | Admix | 84.2±4.1 | 92.5±2.5 | 75.6±4.3 | 75.6±3.9 | 82.0±1.9 | 39.3±4.3 | 33.0±2.8 | 37.0±5.0 | 52.7±4.0 | 40.5±2.3 |
| | Ens | 91.0±1.1 | 82.0±0.4 | 86.2±1.1 | 76.8±0.9 | 84.0±0.7 | 70.2±0.7 | 84.6±0.5 | 70.0±1.3 | 85.3±2.7 | 77.5±0.8 |
| | AdaEA | 88.0±1.7 | 80.2±1.1 | 82.1±1.8 | 71.9±1.5 | 80.6±0.7 | 61.8±1.3 | 79.8±1.7 | 63.0±2.1 | 78.0±2.4 | 70.7±1.2 |
| | DHF | 90.8±0.8 | 89.1±1.4 | 77.0±2.5 | 72.3±1.9 | 82.3±0.9 | 37.0±2.2 | 27.5±2.3 | 25.4±1.8 | 52.0±2.1 | 35.5±1.1 |
| | BFA | 86.2±0.8 | 89.4±0.8 | 70.2±1.0 | 68.2±0.6 | 79.5±0.5 | 34.0±0.7 | 30.7±0.9 | 27.1±0.4 | 50.9±1.0 | 36.6±0.5 |
| | ANDA | 94.8±1.3 | 97.3±0.8 | 77.9±1.2 | 82.5±0.8 | 88.1±0.5 | 42.0±1.8 | 34.6±0.9 | 35.7±1.6 | 49.0±1.8 | 40.3±0.8 |
| | Ours | 98.8±1.4 | 100.0±0.0 | 99.8±0.7 | 96.8±0.9 | 98.7±0.6 | 98.8±1.4 | 95.4±1.4 | 93.3±1.3 | 94.9±1.5 | 95.4±0.6 |
| ImageNet | Admix | 91.4±1.7 | 83.3±2.7 | 81.8±2.7 | 67.3±3.0 | 81.0±1.1 | 21.8±3.4 | 34.3±2.4 | 24.1±3.0 | 31.6±2.9 | 28.0±1.7 |
| | Ens | 69.6±1.1 | 65.9±1.7 | 65.0±2.1 | 52.2±1.2 | 63.2±0.4 | 44.8±2.0 | 66.8±0.7 | 24.9±1.9 | 36.0±1.4 | 43.1±0.5 |
| | AdaEA | 73.8±2.2 | 66.6±2.2 | 61.0±3.0 | 46.4±3.8 | 62.0±1.6 | 36.6±2.2 | 54.2±2.4 | 23.2±1.8 | 28.4±2.7 | 35.6±1.4 |
| | DHF | 98.8±0.6 | 96.2±1.0 | 95.4±1.6 | 88.3±2.0 | 94.7±0.7 | 38.8±2.4 | 52.6±2.8 | 40.1±2.2 | 52.4±2.6 | 46.0±1.5 |
| | BFA | 99.5±0.6 | 97.5±0.6 | 95.5±0.6 | 90.2±1.0 | 95.9±0.4 | 39.0±0.8 | 52.5±0.6 | 44.2±1.7 | 56.2±1.5 | 49.6±0.4 |
| | ANDA | 95.5±0.5 | 87.9±0.6 | 86.6±1.2 | 70.3±0.7 | 85.1±0.5 | 37.9±1.1 | 53.4±0.9 | 35.9±1.5 | 46.8±1.0 | 43.5±0.6 |
| | Ours | 99.9±0.4 | 98.1±1.4 | 97.5±1.4 | 89.3±2.0 | 96.0±1.0 | 45.6±3.4 | 65.9±2.8 | 52.4±2.9 | 65.2±2.4 | 56.9±1.8 |

Table 9: Mean $\pm$ standard deviation of TSRs (%) on ImageNet using ViT-Tiny as the surrogate, except for Ens and AdaEA, which use 2 CNN and 2 ViT surrogates.

| Attack | CNN Models | | | | | ViT Models | | | | |
|---|---|---|---|---|---|---|---|---|---|---|
| | RN-50 | WRN-101 | BiT-50 | BiT-101 | Avg. | ViT-B | DeiT-B | Swin-B | Swin-S | Avg. |
| Ens | 69.6±1.1 | 65.9±1.7 | 65.0±2.1 | 52.2±1.2 | 63.2±0.4 | 44.8±2.0 | 66.8±0.7 | 24.9±1.9 | 36.0±1.4 | 43.1±0.5 |
| AdaEA | 73.8±2.2 | 66.6±2.2 | 61.0±3.0 | 46.4±3.8 | 62.0±1.6 | 36.6±2.2 | 54.2±2.4 | 23.2±1.8 | 28.4±2.7 | 35.6±1.4 |
| Admix | 39.8±2.8 | 48.8±2.4 | 47.4±4.1 | 39.4±2.4 | 43.9±1.6 | 48.6±2.5 | 66.6±2.3 | 23.8±1.9 | 28.4±2.1 | 41.8±1.3 |
| DHF | 43.8±2.5 | 51.6±2.6 | 55.8±3.7 | 45.7±3.5 | 49.2±1.8 | 71.0±2.6 | 79.0±2.3 | 30.0±3.5 | 38.1±4.1 | 54.5±1.8 |
| BFA | 55.0±0.0 | 60.6±0.5 | 63.7±1.0 | 50.6±0.5 | 57.5±0.0 | 75.4±0.5 | 90.6±0.5 | 40.7±1.0 | 46.4±0.5 | 63.3±0.4 |
| ANDA | 55.6±0.8 | 64.5±0.5 | 73.2±0.9 | 62.8±0.4 | 64.0±0.3 | 68.9±0.6 | 78.0±0.2 | 42.6±0.5 | 54.7±0.8 | 61.0±0.2 |
| Ours | 62.7±3.8 | 71.2±4.0 | 73.1±3.0 | 63.4±3.7 | 67.6±2.0 | 82.8±4.8 | 90.5±3.0 | 52.3±4.3 | 63.9±4.2 | 72.4±2.3 |

