# OpenReview forum: "Consensus-Robust Transfer Attacks via Parameter and Representation Perturbations"
_NeurIPS.cc/2025/Conference — NeurIPS 2025 poster_

### Official Review · Reviewer_bSRq · 2025-06-08

**Clarity:** 4
**Significance:** 3
**Originality:** 3
**Rating:** 5
**Confidence:** 4

**Summary:**

The paper introduces CORTA, a novel transfer-based black-box adversarial attack designed to enhance the transferability of adversarial examples across deep neural networks (DNNs). It identifies two primary sources of transfer failure—decision-boundary variation and representation drift—and proposes a unified framework to address these through parameter perturbations and feature blending on a surrogate model. The approach is formalized as a Distributionally Robust Optimization (DRO) problem, with practical first-order approximations for scalability. The authors claim that CORTA outperforms state-of-the-art methods, achieving, for instance, a 19.1% higher transfer success rate (TSR) when transferring from ResNet-18 to Swin-B on CIFAR-100. Extensive experiments on ImageNet and CIFAR-100 across convolutional and transformer architectures are presented to support these claims.

**Questions:**

See weaknesses for details.

**Ethical Concerns:**

["NO or VERY MINOR ethics concerns only"]

**Final Justification:**

The authors addressed my major concerns. I also paid attention to the other reviewers' questions and the replies by the authors. I consider this paper to be of high quality and useful for community development. Therefore I increase my rating by 1 point.

**Limitations:**

Yes

**Quality:**

3

**Strengths And Weaknesses:**

Strengths:
1. The authors provide a rigorous theoretical analysis, deriving upper bounds on the worst-case adversarial loss using Lipschitz continuity and first-order approximations. This enhances the credibility of the proposed approximations for parameter linearization and Monte Carlo sampling for feature blending.

2. The paper's conceptualization of transferability as a consensus-robust optimization problem is innovative. By modeling target models as perturbed versions of the surrogate via parameter and representation channels, it provides a principled approach to tackle both decision-boundary variation and representation drift. The DRO formulation is a strong theoretical contribution, aligning well with the goal of robust transferability.

Weaknesses:
1. The paper explicitly states that it follows prior work in reporting only average results without error bars or statistical significance tests. This is a significant methodological flaw, as it hinders the assessment of result reliability, especially given the variability inherent in adversarial attack experiments. For example, Table 1 and Table 2 report TSRs without confidence intervals, making it difficult to evaluate the consistency of CORTA's performance across runs or datasets.

2. While CORTA is tested with ResNet-18 and ViT-Tiny as surrogates, the baselines Ens and AdaEA use multiple surrogates (2 CNNs and 2 ViTs). This discrepancy makes direct comparisons less fair, as ensemble methods inherently leverage more diverse surrogate information. The paper would benefit from evaluating CORTA with multiple surrogates to better align with these baselines and demonstrate its robustness.

3. This paper briefly mentions hyperparameters (e.g., perturbation probability $\rho=0.5$, blending proportion $\lambda$ sampled from U[0.25,1], and trade-off coefficient $\beta$ tuned empirically), but lacks a detailed ablation study to justify these choices. For instance, the choice of $\lambda_{min}=0.25 for feature blending is not explained, nor is there an analysis of how sensitive CORTA's performance is to variations in $\beta$ or $\rho$. This omission limits the understanding of the method's robustness to hyperparameter settings.

---

> ### Author Rebuttal · Authors · 2025-07-31
>
> **W.1** The paper explicitly states that it follows prior work in reporting only average results without error bars or statistical significance tests. This is a significant methodological flaw, as it hinders the assessment of result reliability, especially given the variability inherent in adversarial attack experiments. For example, Table 1 and Table 2 report TSRs without confidence intervals, making it difficult to evaluate the consistency of CORTA's performance across runs or datasets.
>
> **A.1**  Thank you for highlighting the importance of reporting error bars or statistical significance tests. While prior papers do not include error bars, we appreciate your suggestion, as providing confidence intervals offers a clearer assessment of result reliability—particularly given the inherent variability in adversarial attack experiments.
>
> We re-ran the experiments from Table 1 and Table 2 on 100 randomly selected samples, repeating each experiment 20 times with different random seeds. The results are now reported as mean ± standard deviation. Please note that these are preliminary results; full experimental results will be included in the revised paper.
>
> Table.1 :
>
> | Dataset   | Attack | RN-50    | WRN-101   | BiT-50   | BiT-101  | Avg      | ViT-B    | Deit-B   | Swin-B   | Swin-S   | Avg      |
> |-----------|--------|----------|-----------|----------|----------|----------|----------|----------|----------|----------|----------|
> |           | Admix  | 84.2±4.1 | 92.5±2.5  | 75.6±4.3 | 75.6±3.9 | 82.0±1.9 | 39.3±4.3 | 33.0±2.8 | 37.0±5.0 | 52.7±4.0 | 40.5±2.3 |
> |           | Ens    | 91.0±1.1 | 82.0±0.4  | 86.2±1.1 | 76.8±0.9 | 84.0±0.7 | 70.2±0.7 | 84.6±0.5 | 70.0±1.3 | 85.3±2.7 | 77.5±0.8 |
> |           | AdaEA  | 88.0±1.7 | 80.2±1.1  | 82.1±1.8 | 71.9±1.5 | 80.6±0.7 | 61.8±1.3 | 79.8±1.7 | 63.0±2.1 | 78.0±2.4 | 70.7±1.2 |
> | CIFAR-100 | DHF    | 90.8±0.8 | 89.1±1.4  | 77.0±2.5 | 72.3±1.9 | 82.3±0.9 | 37.0±2.2 | 27.5±2.3 | 25.4±1.8 | 52.0±2.1 | 35.5±1.1 |
> |           | BFA    | 86.2±0.8 | 89.4±0.8  | 70.2±1.0 | 68.2±0.6 | 79.5±0.5 | 34.0±0.7 | 30.7±0.9 | 27.1±0.4 | 50.9±1.0 | 36.6±0.5 |
> |           | ANDA   | 94.8±1.3 | 97.3±0.8  | 77.9±1.2 | 82.5±0.8 | 88.1±0.5 | 42.0±1.8 | 34.6±0.9 | 35.7±1.6 | 49.0±1.8 | 40.3±0.8 |
> |           | Ours   | 98.8±1.4 | 100.0±0.0 | 99.8±0.7 | 96.8±0.9 | 98.7±0.6 | 98.8±1.4 | 95.4±1.4 | 93.3±1.3 | 94.9±1.5 | 95.4±0.6 |
> |           |        |          |           |          |          |          |          |          |          |          |          |
> |           | Admix  | 91.4±1.7 | 83.3±2.7  | 81.8±2.9 | 67.3±3.0 | 81.0±1.1 | 21.8±3.4 | 34.3±2.4 | 24.1±3.0 | 31.6±2.9 | 28.0±1.7 |
> |           | Ens    | 69.6±1.1 | 65.9±1.7  | 65.0±2.1 | 52.2±1.2 | 63.2±0.4 | 44.8±2.0 | 66.8±0.7 | 24.9±1.9 | 36.0±1.4 | 43.1±0.5 |
> | ImageNet  | AdaEA  | 73.8±2.2 | 66.6±2.2  | 61.0±3.0 | 46.4±3.8 | 62.0±1.6 | 36.6±2.2 | 54.2±2.4 | 23.2±1.8 | 28.4±2.7 | 35.6±1.4 |
> |           | DHF    | 98.8±0.6 | 96.2±1.0  | 95.4±1.6 | 88.3±2.0 | 94.7±0.7 | 38.8±2.4 | 52.6±2.8 | 40.1±2.2 | 52.4±2.6 | 46.0±1.5 |
> |           | BFA    | 99.5±0.6 | 97.5±0.6  | 95.5±0.6 | 90.2±1.0 | 95.9±0.4 | 39.0±0.8 | 52.5±0.6 | 44.2±1.7 | 56.2±1.5 | 49.6±0.4 |
> |           | ANDA   | 95.5±0.5 | 87.9±0.6  | 86.6±1.2 | 70.3±0.7 | 85.1±0.5 | 37.9±1.1 | 53.4±0.9 | 35.9±1.5 | 46.8±1.0 | 43.5±0.6 |
> |           | Ours   | 99.9±0.4 | 98.1±1.4  | 97.5±1.4 | 89.3±2.0 | 96.0±1.0 | 45.6±3.4 | 65.9±2.8 | 52.4±2.9 | 65.2±2.4 | 56.9±1.8 |
>
> Table.2:
>
> | Attack | RN-50    | WRN-101  | BiT-50   | BiT-101  | Avg      | ViT-B    | Deit-B   | Swin-B   | Swin-S   | Avg      |
> |--------|----------|----------|----------|----------|----------|----------|----------|----------|----------|----------|
> | Ens    | 69.6±1.1 | 65.9±1.7 | 65.0±2.1 | 52.2±1.2 | 63.2±0.4 | 44.8±2.0 | 66.8±0.7 | 24.9±1.9 | 36.0±1.4 | 43.1±0.5 |
> | AdaEA  | 73.8±2.2 | 66.6±2.2 | 61.0±3.0 | 46.4±3.8 | 62.0±1.6 | 36.6±2.2 | 54.2±2.4 | 23.2±1.8 | 28.4±2.7 | 35.6±1.4 |
> | Admix  | 39.8±2.8 | 48.8±2.4 | 47.4±4.1 | 39.4±2.4 | 43.9±1.6 | 48.6±2.5 | 66.6±2.3 | 23.8±1.9 | 28.4±2.1 | 41.8±1.3 |
> | DHF    | 43.8±2.5 | 51.6±2.6 | 55.8±3.7 | 45.7±3.5 | 49.2±1.8 | 71.0±2.6 | 79.0±2.3 | 30.0±3.5 | 38.1±4.1 | 54.5±1.8 |
> | BFA    | 55.0±0.0 | 60.6±0.5 | 63.7±1.0 | 50.6±0.5 | 57.5±0.0 | 75.4±0.5 | 90.6±0.5 | 40.7±1.0 | 46.4±0.5 | 63.3±0.4 |
> | ANDA   | 55.6±0.8 | 64.5±0.5 | 73.2±0.9 | 62.8±0.4 | 64.0±0.3 | 68.9±0.6 | 78.0±0.2 | 42.6±0.5 | 54.7±0.8 | 61.0±0.2 |
> | Ours   | 62.7±3.8 | 71.2±4.0 | 73.1±3.0 | 63.4±3.2 | 67.6±2.0 | 82.8±4.1 | 90.5±3.0 | 52.3±4.3 | 63.9±4.2 | 72.4±2.3 |
>
>
> **W.2** While CORTA is tested with ResNet-18 and ViT-Tiny as surrogates, the baselines Ens and AdaEA use multiple surrogates (2 CNNs and 2 ViTs). This discrepancy makes direct comparisons less fair, as ensemble methods inherently leverage more diverse surrogate information. The paper would benefit from evaluating CORTA with multiple surrogates to better align with these baselines and demonstrate its robustness.
>
> **A.2** To ensure a fair comparison, we implemented "CORTA-ensemble" using the exact same surrogate set as Ens and AdaEA—ResNet-18, Inception-v3, ViT-Tiny, and DeiT-Tiny. The results on ImageNet are shown below:
>
> | Method          | RN-50 | WRN-101 | BiT-50 | BiT-101 | Avg (CNN) | ViT-B | DeiT-B | Swin-B | Swin-S | Avg (ViT) |
> |-----------------|-------|---------|--------|---------|-----------|-------|--------|--------|--------|-----------|
> | Ours (ResNet-18)    | 98.5  | 95.8    | 95.5   | 92.4    | 95.5      | 47.6  | 63.8   | 54.2   | 64.1   | 57.4      |
> | Ours (ViT-Tiny)    | 63.6  | 70.4    | 74.4   | 68.3    | 69.2      | 77.8  | 87.8   | 50.7   | 58.4   | 68.7      |
> | Ens             | 71.2  | 63.2    | 62.5   | 54.9    | 63.0      | 42.9  | 62.9   | 26.6   | 36.6   | 42.3      |
> | AdaEA           | 73.5  | 61.4    | 59.1   | 50.9    | 61.2      | 36.9  | 53.8   | 25.0   | 33.4   | 37.3      |
> | Ours (Ensemble) | 96.9  | 94.5    | 94.2   | 91.5    | 94.3      | 82.5  | 94.7   | 71.7   | 77.1   | 81.5      |
>
>
> In the identical 4-model ensemble setting, CORTA significantly outperforms both Ens and AdaEA, demonstrating its effectiveness in multi-surrogate scenarios and its ability to further enhance cross-architecture transfer.
>
> Leveraging multiple surrogate models increases transfer success rates across both CNN-based and ViT-based target models. For example, CORTA improves transfer performance to ViT targets by 24.1\% compared to using a single ResNet-18 surrogate. Similarly, compared to a single ViT-Tiny surrogate, transfer performance to CNN and ViT targets increases by 25.1\% and 12.8\%, respectively. This demonstrates that using multiple, diverse surrogates helps bridge the gap in transferability between very different model architectures.
>
> **W.3** This paper briefly mentions hyperparameters (e.g., perturbation probability $\rho = 0.5$, blending proportion $\lambda$ sampled from U[0.25,1], and trade-off coefficient $\beta$ tuned empirically), but lacks a detailed ablation study to justify these choices. For instance, the choice of $\lambda_{min}=0.25$ for feature blending is not explained, nor is the reananalysis of how sensitive CORTA's performance is to variations in $\beta$ or $\rho$. This omission limits the understanding of the method's robustness to hyperparameter settings.
>
> **A.3** In the following, we provide a more comprehensive explanation of the reasoning behind the selection of the key hyperparameters: $\rho$, $\lambda_{min}$, and $\beta$ -- we will revise the paper to better explain our selection of these hyperparameters.
>
> 1. **Choice of $\rho$:** We set $\rho = 0.5$ based on optimization success on the surrogate model. If $\rho$ is too high (e.g., 0.8), the feature perturbation becomes overly strong, disrupting gradient signals and reducing optimization success. Conversely, if $\rho$ is too low (e.g., 0.2), the perturbation is too weak, resulting in limited transferability to the target model. Therefore, we selected $\rho = 0.5$ for **all experiments** to achieve a balanced trade-off.
>
> 2. **Choice of $\lambda_{min}$:** Similarly, setting $\lambda_{min}$ too high or too low degrades optimization success on the surrogate model. We observed stable performance with $\lambda_{min}$ in the range [0.1, 0.3], and thus chose $\lambda_{min} = 0.25$ for **all experiments** in the paper.
>
> 3. **Choice of trade-off coefficient $\beta$:** $\beta$ is chosen to balance the magnitudes of the two optimization losses in the surrogate model, ensuring that their contributions are comparable. Consequently, $\beta$ is not **task- or dataset-specific**, but rather related to the model architecture. For example, we used $\beta = 0.1$ for ResNet-18 on both ImageNet and CIFAR-100. Additionally, our hyperparameter sensitivity analysis (see Figure 1 and Line 297 of the paper) demonstrates that $\beta$ yields stable results within the range [0.01, 0.1].
>
> We address the sensitivity of these hyperparameters in Lines 294–299 of the paper and provide visualizations in Figures 1, 2, and 3 of the paper, which illustrate the effects of varying $\beta$, $\rho$, and $\lambda_{min}$. As shown, CORTA maintains stable performance for $\beta$ in the range [0.01, 0.1], $\rho$ in [0.5, 1], and $\lambda_{min}$ in [0.1, 0.3].
>
> We hope this clarifies our hyperparameter choices and their sensitivity. We will ensure this discussion is included in the revised paper.

---

> > ### Comment · Reviewer_bSRq · 2025-08-06
> >
> > Thank you for your detailed experimental additions and parameter explanations. These have addressed my major concerns. I have already raised my score from 4 to 5 in the final justification.

---

> > > ### Author Response · Authors · 2025-08-06
> > >
> > > Thank you for your positive feedback and for raising the score. We are pleased that our additional experiments and explanations could address your concerns. We sincerely appreciate your time and careful review of our work.

---

### Official Review · Reviewer_Z58m · 2025-06-26

**Clarity:** 3
**Significance:** 2
**Originality:** 3
**Rating:** 4
**Confidence:** 4

**Summary:**

The authors propose a novel method to generate the transferrable black-box adversarial attacks by incorporating the so called distributionally robust optimization technique leading to the following ingredients: 1) $W+\Delta W$ analysis; 2) layer-wise feature perturbations. As a result, they proposed a very competing results on top of both singular and ensemble-based attack methods.

**Questions:**

Questions:
* Why is stochastic feature blending applied when the output size is $\leq 1/16$ of the input (lines 244-245)? No reason behind it, no ablations
  * What about transformers?
* Unclear why the blending probability $\rho=0.5$ is chosen from the ablation study (Figure 2), because it seems that the best is somewhere around 0.8?

**Ethical Concerns:**

["NO or VERY MINOR ethics concerns only"]

**Final Justification:**

**Update** Increasing the score based on authors' comments.

**Limitations:**

yes

**Quality:**

3

**Strengths And Weaknesses:**

Strengths:
* The very fast speed - see Table 5
* Performance on the well-known datasets like CIFAR-100 and Imagenet is significantly higher than of the competitors numbers

Weaknesses:
* Theoretical analysis is quite weak, mainly because:
  * Line 148: "small $\Delta W$" - a very weak hypothesis as the setup is a black-box and targeting any neural network
  * Line 173: a Lipschitz constant $L_{l}$ can be quite huge so the last term in the optimization objective (6) can be much higher than the $||g_{W}(x, \delta)||_{F}$
* No real data (e.g., a comparison table) for the optimization success rate - it was briefly mentioned in lines 308-309, but no the real analysis
* Minor remarks:
  * line 127: better to use another letter than round $L$ for a set of selected layers - because it is also the notation of the loss function
  * line 184: the incorrect superscript 1 in here: $||g_{W}(x, \delta)||_{F}^{1}$
  * line 206: the blending probability $\rho$ should use the different notation - the same letter stands for the upper bound on $|\Delta W|$

[1] Szegedy, Christian, et al. "Intriguing properties of neural networks." arXiv preprint arXiv:1312.6199 (2013).

---

> ### Author Rebuttal · Authors · 2025-07-31
>
> **W.1.1** Line 148: "small $\Delta W$ - a very weak hypothesis as the setup is a black-box and targeting any neural network
>
> **A.1** For $\Delta W$, we lower the upper bound to minimize loss variations across models (see Eq. at line 170 of the paper, which holds for all $\|\Delta W\|_{F} \leq \rho$, note that $\rho$ can be large). This approach addresses the worst-case scenario, ensuring the DRO objective is satisfied even under the most adverse conditions. We have made a minor correction to the equation at line 170 in the revised paper, but the underlying idea remains unchanged.
>
>
> **W.1.2** Line 173: a Lipschitz constant $L_l$ can be quite huge so the last term in the optimization objective (6) can be much higher than the $||g_{W}(x, \delta)||_{F}$
>
> **A.2** The Lipschitz constant $L_{l}$ is a model-dependent term that appears in the upper bound of the optimization objective, but it does not directly participate in the optimization of adversarial samples. It is independent of the gradient term $||g_{W}(x, \delta)||_{F}$, which is minimized during optimization. Therefore, even if $L_l$ is large, CORTA only optimizes components directly related to the perturbation $\delta$, and does not rely on $L_l$. As a result, the magnitude of the Lipschitz constant has no impact on our optimization process.
>
> **W.2** No real data (e.g., a comparison table) for the optimization success rate - it was briefly mentioned in lines 308-309, but no the real analysis.
>
> **A.3**  We present the following comparison table, which shows the optimization success rates of baseline methods on ImageNet using ResNet-18 as the surrogate model:
>
> |                           | Admix | Ens  | AdaEA | DHF  | BFA  | ANDA | Ours |
> |---------------------------|-------|------|-------|------|------|------|------|
> | Optimization Success Rate | 97.0  | 100  | 100   |99.4  | 99.4 | 96.3 | 69.9 |
>
>
> As shown in the table, most baseline methods (Ens, AdaEA, DHF, BFA) achieve optimization success rates close to 100\%, whereas our method (CORTA) achieves a success rate of 69.9\%.
>
> CORTA’s surrogate success rate is lower than that of baseline attacks mainly due to two factors inherent to its DRO formulation: (1) CORTA optimizes two objectives—representation and parameter channels—rather than a single objective, increasing optimization difficulty; and (2) the feature blending operation, which incorporates the original sample’s latent features, can interfere with the adversarial objective and further reduce success rates on the surrogate model.
>
> However, this lower surrogate success rate is not a practical issue. Attackers can simply discard unsuccessful adversarial examples using the surrogate model and retain only those that succeed, which tend to have higher transfer success rates. In practice, this only slightly increases computational cost, as generating the same number of successful examples requires optimizing about 1.43 times more samples (e.g., 100/69.9 compared to attacks with 100\% surrogate success rate).
>
> We will include these updated results and explanations in the revised version.
>
> **W.3** Minor remarks
>
> **A.4** We will make all necessary revisions in accordance with your valuable suggestions.
>
> **Q.1** Why is stochastic feature blending applied when the output size is $\leq 1/16$ of the input (lines 244-245)? No reason behind it, no ablations
> What about transformers?
>
> **A.5**
> We adopted the "output size $\leq 1/16$ of the input" setting to align with the CFM method ([21] in the paper), which is similar to DHF but focusing on targeted adversarial attacks.
>
> However, this constraint is not essential. Our experiments show that applying stochastic feature blending to all layers—using ResNet-18 as the surrogate—still outperforms baseline methods, as shown in the table below:
>
>
> | Blending layer             | RN-50 | WRN-101 | BiT-50 | BiT-101 | Avg(CNN)   | ViT-B | DeiT-B | Swin-B | Swin-S | Avg(ViT)   |
> |----------------------------|-------|---------|--------|---------|------------|-------|--------|--------|--------|------------|
> | $\leq 1/16$ of the input   | 98.5  | 95.8    | 95.5   | 92.4    | 95.5       | 47.6  | 63.8   | 54.2   | 64.1   | 57.4       |
> | All layer                  | 97.2  | 96.0    | 94.1   | 90.8    | 94.5       | 47.4  | 64.3   | 56.0   | 64.0   | 57.9       |
>
> or Transformer architectures, we apply stochastic feature blending to all linear layers. This setting will be clearly explained in the revised version of the paper.
>
> Overall, this parameter is not critical, and we will update its description accordingly in the revision.
>
> **Q.2** Unclear why the blending probability $\rho = 0.5$ is chosen from the ablation study (Figure 2), because it seems that the best is somewhere around 0.8?
>
> **A.6** Thank you for this insightful question. Our choice of $\rho = 0.5$ was based on optimization performance on the surrogate model. When $\rho$ is too high (e.g., 0.8), the feature perturbation can overly disrupt gradient signals, reducing optimization success on the surrogate. Conversely, when $\rho$ is too low (e.g., 0.2), the perturbation is insufficient, limiting transferability. Thus, $\rho = 0.5$ was selected as a balanced value.
>
> While the ablation study indicates that $\rho = 0.8$ achieves better performance on the target model, our choice of $\rho = 0.5$ does not yield the highest transfer success rate on the target. However, as black-box attackers, we cannot tune hyperparameters based on the target model. Therefore, we selected $\rho = 0.5$ based on the optimization success rate on the surrogate model, even though it may not be optimal for the target model.

---

> > ### Comment · Reviewer_Z58m · 2025-08-03
> >
> > Thanks authors for the ablations. I'll increase the score.

---

> > > ### Author Response · Authors · 2025-08-04
> > >
> > > Thank you for your feedback and for increasing the score. We appreciate your time and consideration in reviewing our work.

---

### Official Review · Reviewer_wbsA · 2025-06-30

**Clarity:** 4
**Significance:** 4
**Originality:** 4
**Rating:** 5
**Confidence:** 5

**Summary:**

The paper introduces CORTA, a novel transfer-based black-box adversarial attack that enhances transferability by addressing decision-boundary variation and representation drift. It frames transferability as a distributionally robust optimization problem, modeling target model variability through parameter and representation perturbations on a single surrogate model. CORTA employs lightweight first-order approximations with theoretical guarantees to ensure robust misclassification. Extensive experiments on ImageNet and CIFAR-100 demonstrate CORTA’s superior performance, achieving a 19.1% higher transfer success rate compared to state-of-the-art baselines, including ensemble methods, across diverse architectures like ResNet-18 and Swin-B. The paper’s key contributions include a consensus-robust formulation, dual-channel surrogate modeling, principled optimization, and the CORTA attack itself, setting a new benchmark for black-box adversarial evaluation.

**Questions:**

Given the paper’s strong theoretical foundation and impressive empirical results with CORTA, no glaring issues stand out.

**Ethical Concerns:**

["NO or VERY MINOR ethics concerns only"]

**Final Justification:**

This paper exceeds the standards for acceptance, offering novelty, strong empirical results, and theoretical depth. It is a valuable addition to the literature and is recommended for publication in its current form.

**Limitations:**

yes

**Paper Formatting Concerns:**

No major formatting violations detected.

**Quality:**

4

**Strengths And Weaknesses:**

Strengths
The paper presents a highly innovative approach with CORTA, effectively addressing adversarial transferability by modeling decision-boundary variation and representation drift through a consensus-robust optimization framework. Its dual-channel surrogate modeling, combining parameter and representation perturbations, is both novel and theoretically grounded, with first-order approximations ensuring scalability and provable guarantees. CORTA’s superior empirical performance, achieving higher transfer success rate over state-of-the-art baselines on ImageNet and CIFAR-100, demonstrates its robustness across diverse architectures like ResNet-18 and Swin-B. The clear articulation of contributions and rigorous experimental validation make this work a significant advancement in black-box adversarial attacks.
Weaknesses
No significant weaknesses were identified in the paper. The theoretical analysis is convincing, the methodology is robust, and the experimental results are compelling, leaving little room for critique.

---

> ### Author Rebuttal · Authors · 2025-07-31
>
> Thank you for your thoughtful review and positive feedback. We are glad to hear that the theoretical foundation and empirical results of CORTA were well-received. We appreciate your recognition of the strengths of the paper, and we will continue to refine and expand on the ideas presented in future work.

---

### Official Review · Reviewer_Gq3G · 2025-07-01

**Clarity:** 3
**Significance:** 3
**Originality:** 3
**Rating:** 4
**Confidence:** 3

**Summary:**

The paper introduces CORTA, a consensus-robust transfer attack for adversarial examples in black-box settings. The key innovation lies in modeling two primary sources of transfer failure—decision-boundary variation and representation drift—as parameter and representation perturbations on a surrogate model. The authors formalize transferability as a distributionally robust optimization (DRO) problem over an uncertainty set of plausible target models and provide efficient first-order approximations with theoretical guarantees. Experiments on ImageNet and CIFAR-100 demonstrate that CORTA outperforms state-of-the-art transfer-based attacks, including ensemble methods, across diverse architectures (CNNs and transformers). Notably, CORTA achieves a 19.1% higher transfer success rate than the strongest baseline when transferring from ResNet-18 to Swin-B on CIFAR-100.

**Questions:**

1. How does CORTA’s computational cost scale with model size (e.g., ViT-Large)? Could approximations mitigate this?

2. Optimization Trade-off: Why does CORTA’s surrogate success rate drop compared to baselines? Is this inherent to the DRO formulation?

3. Hyperparameter generalization:
(1) Are the chosen hyperparameters (e.g., β=0.1) task-specific, or can they be generalized? An analysis on additional datasets would help.
(2) eps = 16/255 is quite large in AEs. What are the results for typical, smaller values of eps?

**Ethical Concerns:**

["NO or VERY MINOR ethics concerns only"]

**Final Justification:**

The review is fair. I've responded to authors' rebuttal as well.

**Limitations:**

Yes

**Quality:**

3

**Strengths And Weaknesses:**

Strengths

- Formulation: The dual-channel perturbation framework (parameter and representation) is a new perspective on adversarial transferability. The DRO formulation is theoretically grounded and well-motivated. The first-order approximations (linearization for parameters, Monte Carlo for representation blending) are computationally efficient and supported by provable bounds.
- Experiments: CORTA consistently outperforms baselines, including ensemble methods, across diverse architectures (CNNs, transformers) and datasets (ImageNet, CIFAR-100). It is also effective against defended models (e.g., adversarial training, input transformations) highlight robustness.
- Analysis: The paper provides formal guarantees (Eq. 6) for the worst-case loss upper bounds. Ablation studies validate the contributions of each component (parameter regularization, feature blending).

Weaknesses
- CORTA requires second-order derivatives (per-sample Hessians), which limits batch parallelization. While the authors argue the cost is manageable, this could hinder scalability for very large models.
- CORTA’s optimization success rate on the surrogate (69.9%) is lower than baselines (e.g., ANDA: 96.3%). The authors justify this by emphasizing target-model success, but this trade-off deserves deeper analysis.
- Although the performance is stable within tested ranges (e.g., β∈[0.01,0.1]), the paper does not explore why these ranges work best or how they generalize to other tasks.

---

> ### Author Rebuttal · Authors · 2025-07-31
>
> **Q.1** How does CORTA's computational cost scale with model size (e.g., ViT-Large)? Could approximations mitigate this?
>
> **A.1** The computational cost of CORTA is closely related to the model size. As the model size increases, the time required to optimize each image also increases. The following table illustrates the optimization time per image using various surrogate models on ImageNet:
>
> |         | ResNet-18 | ResNet-50 | ResNet-152 | ViT-Tiny | ViT-Base | ViT-Large |
> | --------| ----------| ----------| -----------| ---------| ---------| ----------|
> | Time(s) | 1.7       | 2.3       | 4.8        | 2.3      | 2.6      | 5.1       |
>
> As shown in the table, the smallest model (ResNet-18) requires only 1.7 seconds of optimization time per image, while the largest model (ViT-Large) requires 5.1 seconds. Notably, even with the ViT-Large model, CORTA’s computational time remains within an acceptable range compared to ensemble methods that utilize multiple smaller models (Ensemble: 5.2s, AdaEA: 18.8s). Furthermore, CORTA achieves high transfer success rates even when using smaller models.
>
> Thank you for the valuable suggestion regarding the use of approximations to mitigate computational costs. Reducing computational overhead is indeed a central focus of our ongoing research. In particular, we are actively exploring gradient approximation techniques to minimize computational expenses. Our goal is to develop a more lightweight approach for generating adversarial examples, while still maintaining high effectiveness.
>
>
> **Q.2** Optimization Trade-off: Why does CORTA's surrogate success rate drop compared to baselines? Is this inherent to the DRO formulation?
>
> **A.2** CORTA’s surrogate success rate is lower than that of baseline attacks mainly due to two factors inherent to its DRO formulation: (1) CORTA optimizes two objectives—representation and parameter channels—rather than a single objective, increasing optimization difficulty; and (2) the feature blending operation, which incorporates the original sample’s latent features, can interfere with the adversarial objective and further reduce success rates on the surrogate model.
>
> However, this lower surrogate success rate is not a practical issue. Attackers can simply discard unsuccessful adversarial examples using the surrogate model and retain only those that succeed, which tend to have higher transfer success rates. In practice, this only slightly increases computational cost, as generating the same number of successful examples requires optimizing about 1.43 times more samples (e.g., 100/69.9 compared to attacks with 100\% surrogate success rate).
>
>
> **Q.3** Hyperparameter generalization: (1) Are the chosen hyperparameters (e.g., $\beta$=0.1) task-specific, or can they be generalized? An analysis on additional datasets would help. (2) eps = 16/255 is quite large in AEs. What are the results for typical, smaller values of eps?
>
> **A.3**  (1) $\beta$ is chosen to balance the magnitudes of the two optimization losses in the surrogate model, ensuring that their contributions are comparable. Consequently, $\beta$ is not **task- or dataset-specific**, but rather related to the model architecture. For example, we used $\beta = 0.1$ for ResNet-18 on both ImageNet and CIFAR-100. Additionally, our hyperparameter sensitivity analysis (see Figure 1 and Line 297 of the paper) demonstrates that $\beta$ yields stable results within the range [0.01, 0.1].
>
> (2) We selected $\epsilon = 16/255$ in our main experiments to align with baseline methods such as Admix, DHF, and BFA, since black-box attacks are considerably more challenging than white-box attacks. To further assess the robustness and generalizability of our method, we also conducted experiments with a smaller, more typical value of $\epsilon$ ($\epsilon = 8/255$).
>
> For these experiments, we used ResNet-18 as the surrogate model on the ImageNet dataset, setting $\epsilon = 8/255$, step size $\alpha = 0.8/255$, and number of iterations $T = 100$. The table below presents the Transfer Success Rate (TSR) of CORTA and various baseline methods under these settings:
>
> | Method      | RN-50 | WRN-101 | BiT-50 | BiT-101 | Avg(CNN)  | ViT-B | Deit-B | Swin-B | Swin-S | Avg(ViT) |
> |-------------|-------|---------|--------|---------|-----------|-------|--------|--------|--------|----------|
> | Admix       | 68.8  | 60.9    | 58.2   | 48.3    | 59.1      | 13.3  | 18.8   | 15.6   | 19.3   | 16.8     |
> | Ens         | 47.0  | 43.3    | 44.5   | 37.2    | 43.0      | 26.7  | 39.7   | 15.5   | 20.9   | 25.7     |
> | AdaEA       | 48.7  | 43.8    | 43.2   | 35.4    | 42.8      | 23.6  | 39.0   | 15.2   | 20.0   | 24.5     |
> | DHF         | 86.8  | 79.2    | 75.5   | 65.5    | 76.8      | 21.3  | 31.2   | 22.6   | 31.5   | 26.7     |
> | BFA         | 87.3  | 80.6    | 70.8   | 63.3    | 75.5      | 14.0  | 23.5   | 20.0   | 26.6   | 21.0     |
> | ANDA        | 77.4  | 65.9    | 64.6   | 52.2    | 65.0      | 17.8  | 27.4   | 19.4   | 22.5   | 21.8     |
> | CORTA(Ours) | 86.9  | 83.8    | 84.3   | 72.6    | 81.9      | 25.3  | 39.2   | 28.2   | 36.5   | 32.3     |
>
> The results show that even with a reduced perturbation budget ($\epsilon = 8/255$), CORTA consistently outperforms baseline methods. This demonstrates that our approach maintains strong attack effectiveness within a smaller perturbation range, further validating the robustness and generalization capability of CORTA.

---

> > ### Comment · Reviewer_Gq3G · 2025-08-06
> >
> > Thank you for your detailed responses. I appreciate the clarifications provided.
> >
> > Given that this paper focuses on decision-boundary variation and representation drift when transferring from surrogate to target models, including CNN-to-transformer transfer, I found and would like to point out a recent related work: "Enabling Heterogeneous Adversarial Transferability via Feature Permutation Attacks" (2025). That work also tackles cross-architecture transfer, and includes MLPs in addition to CNNs and transformers.
> >
> > While your paper and theirs adopt different methodologies, both address similar core challenges. A more principled comparison between the two, for example by highlighting distinctions in assumptions, objectives, and outcomes/performance, could enhance your contribution. I understand this may be beyond the time limit of rebuttal, but I encourage the authors to consider including such a discussion, even briefly, in the camera-ready version (should the paper be accepted). It would broaden the context of your findings and further strengthen the impact of the work.

---

> > > ### Author Response · Authors · 2025-08-07
> > >
> > > Thank you for bringing the Feature Permutation Attack (FPA) to our attention. We were previously unaware of this work. Below, we provide a brief comparison between CORTA and FPA; a more detailed comparison will be included in the revised paper. As the source code for FPA has not been released, we will also attempt to implement FPA ourselves and experimentally compare its performance with ours in the revised version.
> > >
> > > Both FPA and CORTA aim to generate transferable adversarial examples. FPA specifically targets the transfer from CNN-based surrogate models to ViT/MLP-based target models. In contrast, CORTA is designed for general transferability, supporting transfers between any surrogate and target model architectures, including CNN to CNN/ViT, and ViT to CNN/ViT. Furthermore, CORTA enables the use of ensembles comprising various architectures (e.g., CNN + ViT surrogate models) to generate transferable adversarial examples (see A2 for Reviewer bSRq).
> > >
> > > The two approaches are grounded in fundamentally different mechanisms. FPA is a purely heuristic method that relies on permutation at a feature layer of a CNN-based surrogate model to bridge the gap between the local receptive fields of CNNs and the global attention mechanisms of ViTs and MLPs, thereby enhancing transferability from CNN-based surrogates to ViT/MLP targets. In contrast, CORTA adopts a more systematic approach by explicitly modeling the discrepancies between surrogate and target models—specifically, differences in decision boundaries and latent representations. This leads to the formulation of transferable adversarial example generation as a distributionally robust optimization (DRO) problem, which is then simplified into a practical solution with accompanying theoretical analysis, thereby offering stronger theoretical guarantees.
> > >
> > > The experimental results reported in this rebuttal and in the FPA paper offer a preliminary comparison of their performance. Our findings indicate that CORTA achieves higher transferability than FPA when transferring from CNN to CNN and from CNN to ViT. Moreover, CORTA supports two additional scenarios not addressed by FPA: (1) CORTA exhibits substantially higher transferability to ViT targets when using a ViT surrogate model, and (2) when employing an ensemble of CNN and ViT surrogate models, CORTA demonstrates high transferability to both CNN and ViT target models.

---

### Decision · Program_Chairs · 2025-09-17

**Decision:**

Accept (poster)

**Comment:**

This paper proposes
a transfer attack method named *CORTA* (consensus-robust transfer attack)
to mitigate the two factors that hinder transfer attack: decision-boundary variation and representation drift.

The reviewers appreciate the **strengths** of the paper:
(Gq3G) dual-channel perturbation framework is a new perspective, DRO formulation is theoretically grounded, CORTA outperforms baselines, provides formal guarantees,
(wbsA)  effectively addressing adversarial transferability by (dual-channel surrogate modeling) through a consensus-robust optimization framework, novel and theoretically grounded, first-order approximations ensuring scalability and provable guarantees,
(Z58m) fast, significant performance,
(bSRq) rigorous theoretical analysis, innovative conceptualization of transferability,

The reviewers also find the **weaknesses** of the paper:
(Gq3G) second-order derivative, lower success rate on the surrogate than baselines, superior empirical performance,
(Z58m) weak theoretical analysis, no real data,
(bSRq) no error bars, discrepancy between the surrogate models, no ablation study.


**After the discussion**, many concerns seem resolved and clarified(e.g, ImageNet evaluation, error bars, ablation study), and the reviewers agreed upon accepting the paper as its dual-channel framework and DRO formalization are expected to make a clear contribution of the community.